# Risk factors and microbiological features of recurrent *Escherichia coli* bloodstream infections

Yong Chan Kim[1]◉, Heun Choi[2]◉, Young Ah Kim[3]‡*, Yoon Soo Park◉[1]‡*, Young Hee Seo[4], Hyukmin Lee[4], Kyungwon Lee[4,5]

**1** Division of Infectious Disease, Department of Internal Medicine, Yongin Severance Hospital, Yonsei University College of Medicine, Yongin-si, Republic of Korea, **2** Department of Infectious Diseases, National Health Insurance Service Ilsan Hospital, Goyang-si, Republic of Korea, **3** Department of Laboratory Medicine, National Health Insurance Service Ilsan Hospital, Goyang-si, Republic of Korea, **4** Department of Laboratory Medicine and Research Institute of Bacterial Resistance, Yonsei University College of Medicine, Seoul, Republic of Korea, **5** Seoul Clinical Laboratories Academy, Yongin-si, Republic of Korea

◉ These authors contributed equally to this work.
‡ YAK and YSP also contributed equally to this work.
* YAKIM@nhimc.or.kr (YAK); YSPARKOK2@yuhs.ac (YSP)

**Data Availability Statement:** All data generated or analyzed during this study are included in this published article (and its supplementary files).

**Funding:** This work was supported by the National Health Insurance Service Ilsan Hospital grant (No.

## Abstract

Understanding the risk factors and microbiological features in recurrent *Escherichia coli* BSI is helpful for clinicians. Data of patients with *E. coil* BSI from 2017 to 2018 were collected. Antimicrobial resistance rates of *E. coli* were determined. We also identified the ST131 and ESBL genotype to evaluate the molecular epidemiology of *E. coli*. Whole genome sequencing was conducted on the available ESBL-producing *E. coli* samples. Of 808 patients with *E. coli* BSI, 57 (6.31%) experienced recurrence; 29 developed at 4–30 days after initial BSI (early onset recurrence) and 28 at 31–270 days after initial BSI (late onset recurrence). One hundred forty-nine patients with single episode, whose samples were available for determining the molecular epidemiology, were selected for comparison. Vascular catheterization (adjusted odds ratio [aOR], 4.588; 95% confidence interval [CI], 1.049–20.068), ESBL phenotype (aOR, 2.037; 95% CI, 1.037–3.999) and SOFA score ≥9 (aOR, 3.210; 95% CI, 1.359–7.581) were independent risk factors for recurrence. The proportion of ST131 and ESBL genotype was highest in early onset recurrent BSI (41.4% and 41.4%, respectively), from which *E. coil* had the highest resistance rates to most antimicrobial agents. Whole genome sequencing on 27 of ESBL-producing *E. coli* (11 from single episode, 11 from early onset recurrence, and 5 from late onset recurrence) demonstrated that various virulence factors, resistant genes, and plasmid types existed in isolates from all types of BSI. Risk factors contributing to the recurrence and microbiological features of *E. coli* causing recurrent BSI may be helpful for management planning in the clinical setting.

NHIMC2019CR012 to YAK). The funder played no role in the study design, data analysis, or preparation of the manuscript.

**Competing interests:** The authors have declared that no competing interests exist.

## Introduction

*Escherichia coli* is a common gram-negative bacteria that is component of the normal gut microbiota, but this pathogen may also cause blood stream infections (BSI) in humans [1]. *E. coli* has been identified as the leading causative agent in both community and hospital acquired BSI for the last decade [2–4]. The proportion of BSI episodes caused by *E. coli* reached as high as 27.1% over this period with an incidence rate of more than 48.0 per 100,000 people per year in high-income countries [5]. A study in England suggests that BSI caused by *E. coli* is increasing, and reported that there was a 6% increase in incidence between 2012/13 and 2013/14 [2].

*E. coli* is one of the most common pathogens isolated from recurrent BSI [6–8], with the rates of recurrence in these infections ranging from 3.1% [9] to 9.9% [10]. Several factors have been shown to be associated with *E. coli*-based BSI recurrence [6, 7, 9–11], including hematological malignancy, non-urinary tract infection, inadequate antibiotic therapy, and the presence of a vascular catheter. Antibiotic resistance has also been shown to be associated with an increase in recurrent BSI [12]. In particular, the presence of multidrug-resistant extended-spectrum β-lactamase (ESBL)-producing *E. coli* often prevent effective empirical antibiotic therapy in these patients and restrict definite antibiotic therapy for recurrent BSI [13]. In addition, the presence of several virulence factors has been linked to recurrence, as they prevent host-immune reactions resulting in delayed or failed clearance of these pathogens [9].

Recurrence is a significant problem for *E. coli* BSI, which may require re-admission to the hospital thereby adding to the healthcare costs of this condition and potentially increasing the mortality rate of these infections. Therefore, it is important to investigate the prevalence, clinical characteristics, and microbiological features of recurrent BSI caused by *E. coli*. This may be helpful in the planning management and clinical follow-up for these types of infections. However, recent information for the recurrence, especially after the widespread dissemination of ESBL-producing *E. coli*, is limited.

The aim of this study was to describe the current clinical and microbiological characteristics of patients with recurrent *E. coli* BSI in South Korea, where the *E. coli* strains isolated from blood specimens exhibit high rates of resistance to several antimicrobial agents.

## Materials and methods

### Study design and population

We completed a retrospective study of recurrent and single episode *E. coli* BSIs using data collected by the National Health Insurance Service at Ilsan Hospital in South Korea between January 2017 and December 2018. Our study cohort consisted of adult patients (age ≥18 years) who presented at the Ilsan Hospital with positive blood cultures for *E. coli* at least once over the course of a two-year period. These patients were followed up over the 270 days following their initial *E. coli* BSI episode. Patients who died or were transferred to another hospital within three days of their initial diagnosis were excluded, as were patients who were expected to have recurrent bacteremia due to specific infection, such as infective endocarditis, mycotic aneurysm, or endovascular graft infection. All patients were only included once for each BSI episode and the demographic and clinical characteristics for each participant were extracted from their electronic medical records.

### Definition and classification of recurrent BSI

Recurrent BSI was defined as a positive blood culture from a specimen collected 4 days or later after the first infection. Recurrence was subdivided into early onset (4–30 days after initial BSI) and late onset (31–270 days after initial BSI) [12]. Infections were also classified as community-

or hospital-acquired with community-acquired BSI being defined as any infection occurring within 48 h of admission. Infections that developed more than 48 h after admission were defined as hospital-acquired BSI. Comorbidities were defined using the International Classification of Diseases, 10th revision and the Charlson comorbidity index, which predicts the 10-year mortality in patients who may have a range of comorbid conditions [14], was used to quantify a patient's burden of disease. Prior antibiotic use was defined as use of any antimicrobials for ≥ 3 days in the previous 30 days, and the primary focus of each case of BSI was classified according to the Centers for Disease Control and Prevention/National Healthcare Safety Network surveillance criteria [15]. Empirical antibiotics were regarded to be appropriate if administered within one calendar day of culture and the pathogen was susceptible to any of antibiotics used. Appropriate definite antibiotic therapy was defined as the administration of antibiotics based on the susceptibility, within one calendar day of the susceptibility results becoming available. We classified recurrence as a relapse when a strain from the second episode had an identical PFGE pattern to that of the initial episode. Recurrence was considered a reinfection when the PFGE pattern was different between the initial and subsequent episodes.

## Microbiological analysis

Blood cultures was performed using BacT/ALERT FA/NA (BioMerieux Inc., Durham, NC, USA) with the blood bottles being incubated at 37 ˚C for five days. Individual species were identified using a MALDI Biotyper MALDI-TOF MS (Bruker Daltonik, Bremen, Germany), and antimicrobial susceptibility was evaluated using a MicroScan WalkAway plus system (Beckman Coulter, Inc., West Sacramento, CA, USA) and MicroScan Neg Breakpoint Combo Type 44 panel (Siemens Healthcare Diagnostics, Inc., West Sacramento, CA, USA). The susceptibility results were interpreted using the guidelines set out by the Clinical and Laboratory Standards Institute [16].

We performed polymerase chain reaction (PCR)-sequencing of O25b and O16 to determine the sequence type (ST) 131, and a DNA region corresponding to CTX-M, SHV, and TEM was sequenced to determine ESBL genotype. PFGE was performed on eight pairs of late onset recurrence samples and used to distinguish between relapse and reinfection in these patients. The PFGE patterns were analyzed using InfoQuest FP software (Bio-Rad) and used to generate a dendrogram based on the unweighted pair group method, with an arithmetic average (UPGMA) from the Dice coefficient with 1% band position tolerance and 0.5% optimization settings [17].

We evaluated the ST, virulence factor, plasmid type, and antimicrobial resistance genes in each of the ESBL-producing *E. coli* using whole genome sequencing (WGS). WGS was performed on ESBL-producing *E. coli* isolated from patients with recurrent BSI and single episode BSI as follows. The DNA of freshly subcultured isolates was extracted using GenElute™ Bacterial Genomic DNA Kit (Sigma-Aldrich, St. Louis, MO), and 8 μg of input genomic DNA was then used to produce the relevant sequencing libraries. These whole genome libraries were then sequenced using a NextSeq 550 instrument (Illumina, San Diego, CA, USA), and sequences were assembled using Spades (version 3.11.1) and annotated with Prokka (version 1.13.7). The data from the multilocus sequence typing (MLST), plasmid typing, virulence, and resistance gene evaluations were obtained from the center for genomic epidemiology website (www.genomicepidemiology.org) The virulence score was calculated as the sum of all virulence factors, as described by Johnson et al. [18].

## Statistical analysis

The results of the categorical data are presented as numbers with percentages whereas those of the continuous data are reported as the median and IQR. The continuous variables were

compared using the Mann–Whitney U test and the categorical variables were compared using the chi-squared test. Risk factors for recurrent BSI were identified by logistic regression analysis, and significant variables from the univariate analysis were then included in a multivariate logistic regression analysis. Results were considered statistically significant at P < 0.05. All statistical analyses were performed using SPSS 23.0 software (SPSS, Chicago, IL, USA).

### Ethics statement

The study protocol was approved by the Institutional Review Board (NHIMC-2019-01-018) of the National Health Insurance Service at Ilsan Hospital. As this study was retrospective and data were anonymous, the Institutional Review Board waived the requirement for written informed consent from the participants.

## Results

### Clinical characteristics of patients with recurrent and single episode *E. coli* BSI

We identified a total of 808 patients with *E. coli* BSI admitted in the National Health Insurance Service Ilsan Hospital over the evaluation period, with 57 (6.31%) of these patients experiencing at least one recurrent episode. We then narrowed our evaluation cohort to the clinical characteristics of 206 patients, including 57 patients with recurrent BSI and 149 single episode patients whose samples were available for ST 131 sequencing and ESBL genotyping (Table 1). The median age of this cohort was 77 (interquartile range [IQR], 71–84) and 78 (IQR, 67–85) in the recurrent episode and single episode groups, respectively. We did note that there was a slight female sex bias (64.9% in recurrent episode group and 71.1% in single episode group) and that the proportion of liver cirrhosis was higher in the recurrent episode group (N = 6, 10.5%) than in the single episode group (N = 3, 2.0%). The Charlson comorbidity index was also shown to be higher in patients with recurrent BSI (2, IQR, 0–4) when compared to those with single episode BSI (1, IQR, 0–2). Patients who had a vascular catheter were more likely to experience recurrence (N = 6, 10.5% vs N = 3, 2.0%) and *E. coli* with the ESBL phenotype were more frequently isolated from the recurrent samples (N = 24, 42.1% vs N = 37, 24.8%). The sequential organ failure assessment (SOFA) score was also higher in recurrent episode group (4, IQR, 2–8) than in the single episode group (3, IQR, 1–6), but mortality rates did not differ significantly between these groups (P > 0.999).

The 57 recurrent *E. coli* BSI cases were subdivided into early onset recurrence (29) and late onset recurrence (28). The median time to recurrence following the first episode was 6 days (IQR, 4–15.5) and 66.5 days (IQR, 44.3–134.3) in the early onset and late onset recurrence groups, respectively. S1 Table describes the direct comparison of these two groups.

### Risk factors for recurrent *E. coli* BSI

Univariate logistic regression analysis demonstrated that liver cirrhosis, Charlson comorbidity index ≥3, vascular catheter, ESBL phenotype, and SOFA score ≥9 were all associated with recurrent *E. coli* BSI (Table 2). The multivariate analysis identified the vascular catheter (adjusted odds ratio [aOR], 4.588; 95% confidence interval [CI], 1.049–20.068), ESBL phenotype (aOR, 2.037; 95% CI, 1.037–3.999), and SOFA score ≥9 (aOR, 3.210; 95% CI, 1.359–7.581) as independent risk factors for recurrent BSI.

**Table 1. Comparison of patients with recurrent *Escherichia coli* BSI and single episode *E. coli* BSI.**

| Variables | Total (N = 206) | Single episode (N = 149) | Recurrent episode (N = 57) | P |
|---|---|---|---|---|
| Age, years | 78 (70–84) | 78 (67–85) | 77 (71–84) | 0.861 |
| Sex, n (%) | | | | 0.385 |
| Female | 143 (69.42) | 106 (71.1) | 37 (64.9) | |
| Male | 63 (30.6) | 43 (28.9) | 20 (35.1) | |
| Hospital acquired infection, n (%) | 90 (43.7) | 64 (43.0) | 26 (45.6) | 0.731 |
| Transferred case, n (%) | | | | 0.894 |
| Non-transferred | 114 (55.3) | 83 (55.7) | 31 (54.4) | |
| Long-term care facility | 17 (8.3) | 13 (8.7) | 4 (7.0) | |
| Acute care hospital | 37 (18.0) | 25 (16.8) | 12 (21.1) | |
| Clinic | 38 (18.5) | 28 (18.8) | 10 (17.5) | |
| Comorbidity, n (%) | | | | |
| Diabetes mellitus | 56 (27.2) | 41 (27.5) | 15 (26.3) | 0.862 |
| Heart failure | 8 (3.9) | 6 (4.0) | 2 (3.5) | >.999 |
| Pulmonary disease | 4 (1.9) | 3 (2.0) | 1 (1.8) | >.999 |
| Chronic kidney disease | 23 (11.2) | 15 (10.1) | 8 (14.0) | 0.419 |
| Liver cirrhosis | 9 (4.4) | 3 (2.0) | 6 (10.5) | 0.015 |
| Malignancy | 40 (19.4) | 25 (16.8) | 15 (26.3) | 0.122 |
| Charlson comorbidity index | 1 (0–3) | 1 (0–2) | 2 (0–4) | 0.013 |
| Primary focus of bacteremia, n (%) | | | | 0.114 |
| Non-urinary tract infection | 66 (32.0) | 43 (28.9) | 23 (40.4) | |
| Urinary tract infection | 140 (68.0) | 106 (71.1) | 34 (59.7) | |
| Polymicrobial blood stream infection, n (%) | 11 (5.3) | 9 (6.0) | 2 (3.5) | 0.731 |
| Prior antibiotic use, n (%) | 180 (87.4) | 132 (88.6) | 48 (84.2) | 0.397 |
| Appropriate empirical antibiotic use, n (%) | 152 (73.8) | 112 (75.2) | 40 (70.2) | 0.466 |
| Appropriate definite antibiotic use, n (%) | 183 (88.8) | 131 (87.9) | 52 (91.2) | 0.5 |
| Duration of definite antibiotic use, days | 14 (12–16) | 14 (11–15) | 14 (13–17) | 0.126 |
| Indwelling catheter in the previous 30 days, n (%) | | | | |
| Vascular catheter | 9 (4.4) | 3 (2.0) | 6 (10.7) | 0.014 |
| Urinary catheter | 128 (62.1) | 95 (63.8) | 33 (57.9) | 0.438 |
| *E. coli* with ESBL phenotype, n (%) | 61 (29.6) | 37 (24.8) | 24 (42.1) | 0.015 |
| SOFA score | 3 (1–7) | 3 (1–6) | 4 (2–8) | 0.035 |
| 30-day mortality, n (%) | 13 (6.3) | 10 (6.7) | 3 (5.3) | >.999 |

BSI, bloodstream infection; ESBL, extended spectrum β-lactamase; SOFA, sequential organ failure assessment

## Molecular epidemiology and antimicrobial resistance of *E. coli* strains

Evaluation of the 206 isolates tested revealed that 53 (25.7%) could be classified as ST131, comprising ST131–025 (N = 47) and ST131–016 (N = 6) (Table 3). The ST131–025 strains were more frequently observed in the early onset group (N = 12, 41.4%) when compared to the late onset recurrent episode (N = 6, 21.4%) and single episode (N = 29, 19.5%) groups (*P* = 0.036). *E. coli* with an ESBL genotype were also more commonly observed in the early onset recurrent episode (N = 12, 41.4%) group when compared with either the late onset recurrent episode (N = 7, 25%) or single episode BSI (N = 25, 16.8%) groups (*P* = 0.011). CTX-M-15 (N = 25, 56.8%) was the most common ESBL genotype, followed by CTX-M-14 (N = 11, 25%).

E. coli isolated from the early onset recurrent BSI group had higher resistance rates to most antimicrobial agents than those isolated from either the late onset recurrent BSI or single

**Table 2. Risk factors for recurrent *Escherichia coli* BSI.**

| Variables | Univariate analysis | | Multivariate analysis | |
|---|---|---|---|---|
| | OR (95% CI) | *P* | Adjusted OR (95% CI) | *P* |
| Age≥ 65 | 1.577 (0.703–3.538) | 0.269 | 1.579 (0.663–3.763) | 0.302 |
| Sex | | | | |
| Female | Reference | | Reference | |
| Male | 1.332 (0.696–2.550) | 0.386 | 1.098 (0.533–2.265) | 0.799 |
| Hospital acquired infection | 1.114 (0.603–2.058) | 0.731 | | |
| Transferred case | | | | |
| Non-transferred | Reference | | | |
| Long-term care facility | 0.824 (0.25–2.719) | 0.751 | | |
| Acute care hospital | 1.285 (0.576–2.867) | 0.54 | | |
| Clinic | 0.956 (0.416–2.196) | 0.916 | | |
| Comorbidity | | | | |
| Diabetes mellitus | 0.941 (0.472–1.877) | 0.862 | | |
| Heart failure | 0.867 (0.17–4.424) | 0.863 | | |
| Pulmonary disease | 0.869 (0.089–8.531) | 0.904 | | |
| Chronic kidney disease | 1.459 (0.582–3.654) | 0.42 | | |
| Liver cirrhosis | 5.725 (1.381–23.735) | 0.016 | | |
| Cancer | 1.772 (0.854–3.675) | 0.124 | | |
| Charlson comorbidity index≥ 3 | 1.973 (1.028–3.788) | 0.041 | 1.591 (0.791–3.199) | 0.193 |
| Primary focus of bacteremia | | | | |
| Non-urinary tract infection | Reference | | | |
| Urinary tract infection | 0.6 (0.317–1.134) | 0.116 | | |
| Polymicrobial bloodstream infection | 0.566 (0.118–2.702) | 0.475 | | |
| Antibiotic use ≤1 month before episode | 1.456 (0.608–3.485) | 0.399 | | |
| Appropriate empirical antibiotic use | 0.777 (0.394–1.532) | 0.467 | | |
| Appropriate definite antibiotic use | 1.429 (0.504–4.049) | 0.502 | | |
| Duration of definite antibiotic use | 1.021 (0.995–1.047) | 0.109 | | |
| Indwelling catheter in 30 days | | | | |
| Vascular catheter | 5.84 (1.408–24.222) | 0.015 | 4.588 (1.049–20.068) | 0.043 |
| Urinary catheter | 0.782 (0.419–1.457) | 0.438 | | |
| Tracheal tube | 0.867 (0.17–4.424) | 0.863 | | |
| Nasogastric tube | 1.456 (0.608–3.485) | 0.399 | | |
| *E. coli* with ESBL phenotype | 2.201 (1.156–4.191) | 0.016 | 2.037 (1.037–3.999) | 0.039 |
| SOFA score ≥9 | 3.406 (1.487–7.804) | 0.004 | 3.210 (1.359–7.581) | 0.008 |

BSI, bloodstream infection; OR, odd ratio; CI, confidence interval, ESBL, extended spectrum β-lactamase; SOFA, sequential organ failure assessment

episode BSI groups (Fig 1). Resistance rates to piperacillin/tazobactam, cefoxitin, minocycline, chloramphenicol, fosfomycin, and tigecycline were all very low in all three groups. There were no amikacin-, colistin-, or carbapenem-resistant strains detected in this study.

## WGS of ESBL-producing *E. coli*

Our cohort included a total of 27 ESBL-producing *E. coli* strains suitable for whole genome sequencing (WGS); 11 from single episode samples, 11 from early onset recurrent episode samples, and 5 from late onset recurrent episode samples. The most frequently noted ST in WGS of ESBL-producing *E. coli* was ST131, regardless of BSI episode type (S2 Table). Various

**Table 3. The molecular epidemiology of the *Escherichia coli* specimens isolated from patients with BSI.**

| | Single episode BSI (N = 149) | Early onset recurrent BSI (N = 29) | Late onset recurrent BSI (N = 28) | P |
|---|---|---|---|---|
| ST131, n (%) | 34 (22.8) | 12 (41.4) | 7 (25) | 0.112 |
| O25 | 29 (19.5) | 12 (41.4) | 6 (21.4) | 0.036 |
| O16 | 5 (3.4) | 0 | 1 (3.6) | 0.832 |
| ESBL genotype, n (%) | 25 (16.8) | 12 (41.4) | 7 (25) | 0.011 |
| CTX-M-3 | 0 | 0 | 1 (3.6) | 0.136 |
| CTX-M-14 | 5 (3.4) | 3 (10.3) | 3 (10.7) | 0.078 |
| CTX-M-15 | 17 (11.4) | 5 (17.2) | 3 (10.7) | 0.617 |
| CTX-M-27 | 2 (1.3) | 4 (13.8) | 0 | 0.012 |
| CTX-M-55 | 1 (0.7) | 0 | 0 | >0.999 |

BSI, bloodstream infection; ST, sequence type; ESBL, extended spectrum β-lactamase

antimicrobial resistance genes and plasmid types were detected in each of the isolates from each of the different BSI groups, but there was no dominant resistance gene or plasmid type. We also detected a wide range of virulence genes. The median virulence scores were 5 (IQR, 3–7), 4.5 (IQR, 1–6), and 5 (IQR, 2–8) for early onset recurrent, late onset recurrent, and single episode BSI, respectively.

## Relapses versus reinfections in late onset recurrent BSI

We performed PFGE with the available 107 isolates, which consisted of early onset recurrent cases (n = 25), the first (n = 24) and second (n = 8) isolates of late onset recurrent cases, and

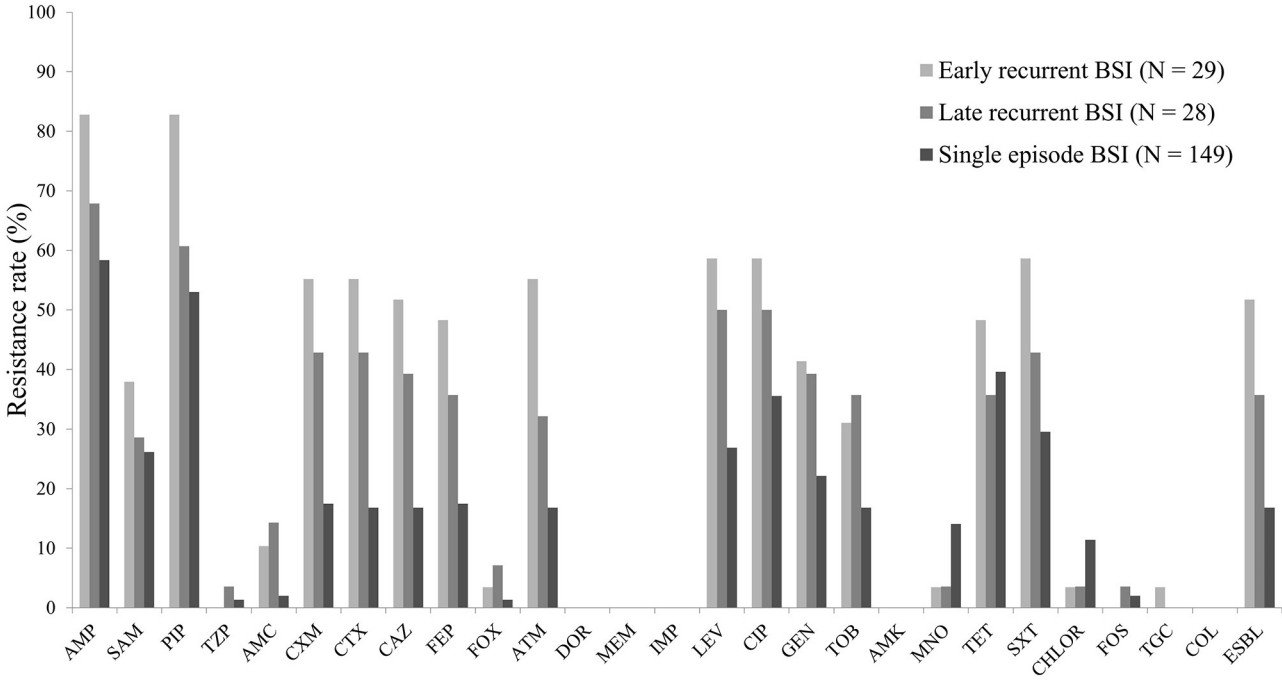

**Fig 1. Antimicrobial resistance rates of the *Escherichia coli* specimens isolated from patients with bloodstream infections (N = 206).**
Abbreviations: BSI, bloodstream infection; Amp, ampicillin; Sam, ampicillin/sulbactam; Pip, piperacillin; Tzp, piperacillin/tazobactam; Amc, amoxicillin/clavulanic acid; Cxm, cefuroxime; Ctx, cefotaxime; Caz, ceftazidime; Fep, cefepime; Fox, cefoxitin; Atm, aztreonam; Dor, doripenem; Mem, meropenem; Imp, imipenem; Lev, levofloxacin; Cip, ciproflroxacin; Gen, gentamicin; Tob, tobramycin; Amk, amikacin; Mno, minocycline; Tet, tetracycline; Sxt, cotrimoxazole; Chlor, chloramphenicol; Fos, fosfomycin; Tgc, tigecycline; Col, colistin; ESBL, extended spectrum β-lactamase.

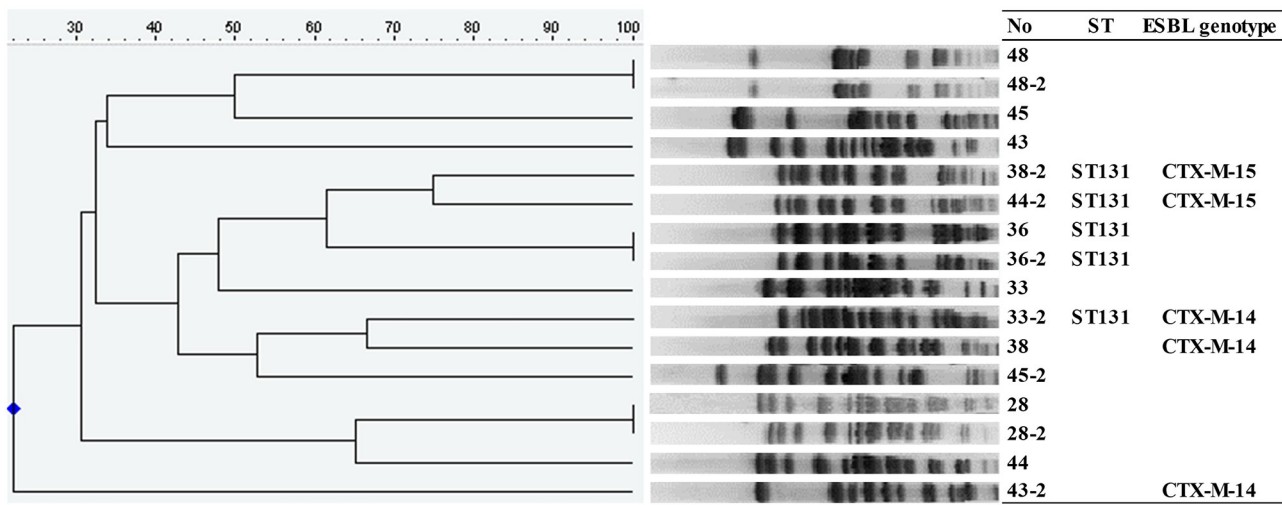

**Fig 2. Dendrogram of XbaI-mediated restriction mapping of the DNA from the *Escherichia coli* specimens isolated from late onset recurrent bloodstream infections (N = 16).** Pulsed-field gel electrophoresis was performed using a Lambda Ladder (Promega, Fitchburg, WI, USA). Abbreviations: ST, sequence type; ESBL, Extended spectrum β-lactamase.

single episode BSI cases (n = 50) (S1 Fig). Among eight paired isolates of the initial and subsequent BSI episodes from the late onset recurrent BSI group, three pairs (28/28-2, 36/36-2, and 48/48-2) of isolates had identical PFGE patterns, ST type, and ESBL genotype, which suggests that these patients were suffering from relapse rather than reinfection (Fig 2).

## Discussion

This study revealed that recurrent *E. coli* BSI was not uncommon in this cohort and that the rate of early onset recurrence was similar to that of late onset recurrence. Our evaluations also identified several independent risk factors for recurrent *E. coli* BSI including the presence of a vascular catheter, ESBL phenotype, and SOFA score. In addition, molecular evaluation of these samples revealed that multidrug-resistant *E. coli* ST131 strains, which produce ESBL, were more common in early onset recurrent BSI than in late onset recurrent BSI or single episode BSI. PFGE analysis suggested that some of the cases in the late onset recurrent BSI group could be attributed to relapse rather than reinfection. WGS revealed that ESBL-producing *E. coli* strains were characterized by a variety of resistance genes and plasmid types.

Recurrence is a serious complication associated with BSI with the rate of recurrence often being dependent on the causative pathogen, definition of recurrence, and follow-up duration. However, recurrence rates for several common pathogens have already been established, these include *Staphylococcus aureus* (5–12%) [19–21], *Streptococcus pneumonia* (2.3%) [22], *Acinetobacter baumannii* (5.6%) [23], and *Salmonella* species (15.7%) [24]. We found that 6.3% of the patients with *E. coli* BSI evaluated in this study experienced some recurrence of their infection. Our study also defined the recurrence rate for two specific subgroups of these patients, early onset (3.6%) and late onset (3.5%) recurrences. To date, there is limited information describing changes in the recurrence rate in response to time following the first episode, which may be clinically relevant and could help plan an appropriate follow-up for *E. coli* BSI patients.

We demonstrated that ESBL phenotype was an important risk factor for recurrent BSI. Maslow et al. showed that impaired host defense responses were closely associated with recurrent episodes [9]. A study by Sanz-García et al. suggested that male sex, the presence of

hematological malignancy, inadequate antibiotic use, and extra-urinary infections were all independent risk factors for recurrence [11]. As these studies were conducted before the widespread dissemination of ESBL-producing *E. coli* [25–27], they could not evaluate the association between ESBL phenotype and recurrence. Patients with BSI caused by ESBL-producing *E. coli* may have an increased risk for subsequent recurrent BSI due to the multidrug-resistant traits of these bacteria. There is some evidence to support this hypothesis, as other studies have shown that antibiotic resistance observed in the initial *E. coli* infection increases the risk of recurrent BSI [12, 13].

Recurrent BSI can be caused by infection with identical (relapse) or different (reinfection) isolates, usually determined by molecular typing. PFGE analysis was used to evaluate these two types of recurrences in the available late onset samples. Although relapse was less common than reinfection, it was still observed at a substantial rate. Reinfection is associated with a higher degree of severity of illness and may be difficult to modify and/or prevent [6]. In contrast, relapse may occur from the remnant foci of the initial infection which are produced following the undertreatment of the original BSI. This implies that appropriate therapy, such as the elimination of any residual infection foci, may reduce the rate of relapse.

ESBL-producing *E. coli* BSI has increased over the past decade owing to the widespread dissemination of the ST 131 clone containing CTX-M-15 [25–27]. To date, limited data are available for the rate of ESBL-producing *E. coli* in recurrent BSI. We showed that as much as 33.3% of our recurrent cases were the result of ESBL-producing *E. coli* and that CTX-M-15 was the most common ESBL genotype in these patients. Multiple virulence factors may be involved in the survival of ESBL-producing *E. coli* and our data suggests that *gad*, which is associated with the protection of cells in acidic environments, was the most prevalent virulence gene in our samples. We also identified strains positive for *iss*, *iha*, and *sat*, which are involved in immune evasion, adhesion, and protease production, respectively [28, 29].

Although our study provides some much-needed insight into recurrent BSI, it does have some limitations. First, not all of the *E. coli* BSI isolates were included in the analysis; therefore, there may have been a selection bias. Second, we could not perform PFGE on the *E. coli* isolates from patients with early onset recurrent BSI. However, we postulated that most of the early onset recurrent BSIs were infected by identical strains. *E. coli* isolated from early onset recurrent BSI was highly resistant to third-generation cephalosporin or ciprofloxacin, which are most commonly used as initial therapy after the identification of gram-negative BSI. This means that the initial empirical antibiotic therapy was likely ineffective, therefore, it is reasonable to assume that the majority of the early onset recurrence cases might be caused by remnants of their initial infection. Third, we could not determine the effect of antimicrobial resistance genes, plasmid types, and virulence factors detected in ESBL-producing *E. coli* isolates on recurrent BSI due to limited numbers of isolates included in the WGS test. This limitation precluded comprehensive and accurate explanation for genetic characteristics of *E. coli* causing recurrent BSI. Further studies are needed to provide more information on molecular characteristics of ESBL-producing *E. coli* causing BSI. Lastly, we showed that the presence of a vascular catheter was associated with recurrent *E. coli* BSI, which agrees with the findings of previous studies [13, 30]. However, the number of patients with a vascular catheter was too low to draw any conclusive statements. Therefore, the findings related to those few patients will need to be interpreted with caution.

## Conclusions

Our evaluations identified several independent risk factors for recurrent *E. coli* BSI, which may be helpful to healthcare providers when planning patient follow-up after initial *E. coli* BSI.

The detection of variable molecular feature in the *E. coli* strains responsible for these recurrent infections suggests the need for an in-depth study into these strains to identify the microbiological determinants associated with recurrent *E. coli* BSI.

## Supporting information

**S1 Fig. Dendrogram of XbaI-mediated restriction mapping of the DNA from the *Escherichia coli* specimens isolated from bloodstream infections (N = 107).** Pulsed-field gel electrophoresis was performed using a Lambda Ladder (Promega, Fitchburg, WI, USA).
(TIF)

**S1 Table. Comparison of patients with early- and late-onset recurrent *Escherichia coli* BSIs.**
(DOCX)

**S2 Table. Whole genome sequencing data of ESBL-producing *E. coli* isolated from patients with BSI.**
(DOCX)

## Acknowledgments

We thank Jung Hwa Hong, a member of research institute at the National Health Insurance Service Ilsan Hospital.

## Author Contributions

**Conceptualization:** Young Ah Kim, Yoon Soo Park.

**Data curation:** Heun Choi, Young Ah Kim, Young Hee Seo.

**Formal analysis:** Yong Chan Kim, Young Ah Kim.

**Supervision:** Young Ah Kim, Yoon Soo Park.

**Writing – original draft:** Yong Chan Kim.

**Writing – review & editing:** Heun Choi, Yoon Soo Park, Hyukmin Lee, Kyungwon Lee.

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
