## [Decision Letter · Decision Letter 0]

18 Aug 2022

PONE-D-22-18648Risk factors and microbiological features of recurrent Escherichia coli bloodstream infectionsPLOS ONE

Dear Dr. Park,

Thank you for submitting your manuscript to PLOS ONE. After careful consideration, we feel that it has merit but does not fully meet PLOS ONE’s publication criteria as it currently stands. Therefore, we invite you to submit a revised version of the manuscript that addresses the points raised during the review process.

Please address comments on the manuscript and improve presentation of Table 1. 

We look forward to receiving your revised manuscript.

Kind regards,

Iddya Karunasagar

Academic Editor

PLOS ONE

Journal Requirements:

Additional Editor Comments:

The reviewer has many important comments to improve the manuscript. Please revise considering all comments point by point.

Reviewers' comments:

Reviewer's Responses to Questions

**Comments to the Author**

1. Is the manuscript technically sound, and do the data support the conclusions?

Reviewer #1: Yes

2. Has the statistical analysis been performed appropriately and rigorously? 

Reviewer #1: No

3. Have the authors made all data underlying the findings in their manuscript fully available?

Reviewer #1: Yes

4. Is the manuscript presented in an intelligible fashion and written in standard English?

Reviewer #1: Yes

5. Review Comments to the Author

Reviewer #1: The authors hve carried out a retrospective analysis of Esch.coli BSI with recurren and re-infection and have compared resistance and virulence genes.

As it is a retrospective study, details of how the folow up data was collected needs to be exlained.s

No where have the authors juxtaposed the different resistance or virulence genes among the groups,to compare. with an appropriate statistical test.

Table 1. just lists the various genes. This may be modified to a comparative table.

Minor points have been highlighted in the attached manuscript.

6. PLOS authors have the option to publish the peer review history of their article (what does this mean?). If published, this will include your full peer review and any attached files.

Reviewer #1: **Yes: **Reba Kanungo MD,PhD Former Dean Research & HoD Microbiology ,PIMS,Puducherry, India

---

## [Author Response · Author response to Decision Letter 0]

16 Sep 2022

To editor, 

Thank you for your constructive comments and criticism on our manuscript. The manuscript has been carefully rechecked and appropriate changes have been made in accordance with the journal requirements and the reviewers’ suggestions. We believe that the quality of the manuscript has significantly improved after incorporating the reviewers’ suggestions. Our point-by-point responses to the reviewers are provided in the “Response to reviewers” file. Furthermore, we have ensured that the manuscript meets PLOS ONE's style requirements. We have now provided the original (uncropped and unadjusted) images of the underlying gel results reported in Fig 2, as S1 Fig.

We hope that the revised manuscript is now suitable for publication in your journal. We look forward to your response.

Response to Reviewer

We thank the reviewer for their thoughtful comments and suggestions, which have enhanced the manuscript and produced a balanced account of the research.

Reviewer #1:

Comment #1

As it is a retrospective study, details of how the follow up data was collected needs to be explained.

Authors’ response to comment #1>

To address your comment. The Department of Laboratory Medicine in Ilsan Hospital, where this study was conducted, has been collecting E. coli strains that caused bloodstream infection for research. Since they were not specimen of human origin, informed consent was not required from the subjects for collection. Patients’ information related to E. coli strains was collected retrospectively through medical records.

Comment #2

No where have the authors juxtaposed the different resistance or virulence genes among the groups,to compare. with an appropriate statistical test.

Table 1. just lists the various genes. This may be modified to a comparative table.

Authors’ response to comment #2>

Please verify whether your comment is about Table 4 and not Table 1. If you are referring to Table 4, we could not compare antimicrobial resistance genes, plasmid types, and virulence factors detected in each of the ESBL-producing E. coli isolates from each of the different BSI groups due to limited numbers of isolates included in the WGS test. 

Therefore, we decided to present Table 4 as supplementary data (S2 Table). We further describe this limitation in the “Discussion” section, as follows (Page 17, Line 322–326):

“Third, we could not compare antimicrobial resistance genes, plasmid types, and virulence factors detected in each of the ESBL-producing E. coli isolates from each of the different BSI groups due to limited numbers of isolates included in the WGS test. Further studies are needed to provide more information on molecular characteristics of ESBL-producing E. coli causing BSI.”

Comment #3

 Minor points have been highlighted in the attached manuscript.

-There is no shortening in “Short Title”

Authors’ response to comment #3>

We agree and we have revised the short title, as follows: 

“Recurrent Escherichia coli bloodstream infections”

Comment #4

 Minor points have been highlighted in the attached manuscript.

-Grammar error in “Abstract”

Authors’ response to comment #4>

We concur and we have corrected the sentence as follows (Page 3 Line 31–32): 

Original: “We also identify the ST131 and ESBL genotype to…”

-> Revised: “We also identified the ST131 and ESBL genotype to…”

Comment #5

 Minor points have been highlighted in the attached manuscript.

-What statistical test was applied to note significant differences in Table 3?

Authors’ response to comment #5>

We acknowledge your detailed comment. To address your comment, we performed a statistical test. 

We also modified the manuscript according to the statistical test result (Page 12, Line 210–215).

Original: “ST131 strains were more frequently observed in the early onset group (N = 12, 41.4%) when compared to the late onset recurrent episode (N = 7, 25.0%) and single episode (N = 34, 22.8%) groups. E. coli with an ESBL genotype were also more commonly observed in the early onset recurrent episode (N = 12, 41.4%) group when compared with either the late onset recurrent episode (N = 7, 25%) or single episode BSI (N = 25, 16.8%) groups.”

-> Revised: ‘ST131–025 strains were more frequently observed in the early onset group (N = 12, 41.4%) when compared to the late onset recurrent episode (N = 6, 21.4%) and single episode (N = 29, 19.5%) groups (P = 0.036). E. coli with an ESBL genotype were also more commonly observed in the early onset recurrent episode (N = 12, 41.4%) group when compared with either the late onset recurrent episode (N = 7, 25%) or single episode BSI (N = 25, 16.8%) groups (P = 0.011).”

---

## [Decision Letter · Decision Letter 1]

25 Oct 2022

PONE-D-22-18648R1Risk factors and microbiological features of recurrent Escherichia coli bloodstream infectionsPLOS ONE

Dear Dr. Park,

Thank you for submitting your manuscript to PLOS ONE. After careful consideration, we feel that it has merit but does not fully meet PLOS ONE’s publication criteria as it currently stands. Therefore, we invite you to submit a revised version of the manuscript that addresses the points raised during the review process.

Please provide clarifications for patient consent and add limitation of the study. 

We look forward to receiving your revised manuscript.

Kind regards,

Iddya Karunasagar

Academic Editor

PLOS ONE

Journal Requirements:

Additional Editor Comments:

Please clarify regarding patient consent and add a note on the limitation of the study.

Reviewers' comments:

Reviewer's Responses to Questions

**Comments to the Author**

1. If the authors have adequately addressed your comments raised in a previous round of review and you feel that this manuscript is now acceptable for publication, you may indicate that here to bypass the “Comments to the Author” section, enter your conflict of interest statement in the “Confidential to Editor” section, and submit your "Accept" recommendation.

Reviewer #1: All comments have been addressed

2. Is the manuscript technically sound, and do the data support the conclusions?

Reviewer #1: Partly

3. Has the statistical analysis been performed appropriately and rigorously? 

Reviewer #1: Yes

4. Have the authors made all data underlying the findings in their manuscript fully available?

Reviewer #1: Yes

5. Is the manuscript presented in an intelligible fashion and written in standard English?

Reviewer #1: Yes

6. Review Comments to the Author

Reviewer #1: Minor revision is required with regards to the following

1. E. coli strains that caused bloodstream infection for research. Since they were not specimen of human origin, informed consent was not required from the subjects for collection� How did this reflect BS infections in specific patients ?

However the following stamen is contradictory: Our study cohort consisted of adult patients (age ≥18 92 years) who presented at the Ilsan Hospital with positive blood cultures for E. coli at least 93 once over the course of a two-year period.

Was any permission sought and granted from the hospital authorities to use patient data? If so this must be mentioned.

2. The paper still lacks in clarity with regards to the genetic studies on resistance and recurrence outcome . This may be highlighted .

7. PLOS authors have the option to publish the peer review history of their article (what does this mean?). If published, this will include your full peer review and any attached files.

Reviewer #1: No

---

## [Author Response · Author response to Decision Letter 1]

18 Nov 2022

We thank the editor and reviewers for their thoughtful comments and suggestions, which have enhanced the manuscript and produced a balanced account of the research.

Journal Requirements:

Authors’ response>

We have reviewed and changed the reference list. We found that reference numbers 12 and 14 were duplicated, so we corrected the reference number after deleting one of them. These changes were also applied to the introduction, methods, results, and discussion. 

Additional Editor Comments:

Please clarify regarding patient consent and add a note on the limitation of the study.

Authors’ response>

In response to the reviewer’s comments, statements regarding patient consent and descriptions of the limitation of this study are listed below.

Reviewer #1:

Comment #1

E. coli strains that caused bloodstream infection for research. Since they were not specimen of human origin, informed consent was not required from the subjects for collection� How did this reflect BS infections in specific patients ?

However the following stamen is contradictory: Our study cohort consisted of adult patients (age ≥18 92 years) who presented at the Ilsan Hospital with positive blood cultures for E. coli at least 93 once over the course of a two-year period.

Was any permission sought and granted from the hospital authorities to use patient data? If so this must be mentioned.

Authors’ response to comment #1>

We regret to inform you that there were errors and omissions in our prior explanation.

E. coli strains identified in blood culture were of human origin. However, since bacteria isolated from the human body do not contain the genetic information of humans, there are no restrictions on the collection or use of the collected strains. This matter is addressed in detail in the bylaws of the Institutional Review Board (by which this study was approved). Young Ah Kim (Department of Laboratory Medicine at Ilsan Hospital) harvested several strains isolated from blood culture, including E. coli for another study. E. coli strains analyzed for this study were already collected and matched with the ID number of patients, who had experienced bloodstream infections (BSIs) due to corresponding strains. Therefore, clinical information on patients with BSIs caused by a specific strain could likewise be obtained through a medical record review. The aforementioned information was disclosed when receiving the Institutional Review Board approval for this study.

Comment #2

The paper still lacks in clarity with regards to the genetic studies on resistance and recurrence outcome. This may be highlighted.

Authors’ response to comment #2>

We attempted to evaluate the genetic characteristics of ESBL-producing E. coli strains from each of the different BSI groups and determine the effects of genetic factors on recurrence. However, we could not determine the effect of antimicrobial resistance genes, plasmid types, and virulence factors detected in ESBL-producing E. coli isolates on recurrence due to the limited number of isolates included in the WGS test. 

We further describe this limitation in the “Discussion” section of the paper (Page 17, Line 322–326):

“Third, we could not determine the effect of antimicrobial resistance genes, plasmid types, and virulence factors detected in ESBL-producing E. coli isolates on recurrent BSI due to limited numbers of isolates included in the WGS test. This limitation precluded comprehensive and accurate explanation for genetic characteristics of E. coli causing recurrent BSI.”

---

## [Decision Letter · Decision Letter 2]

22 Dec 2022

Risk factors and microbiological features of recurrent Escherichia coli bloodstream infections

PONE-D-22-18648R2

Dear Dr. Park,

We’re pleased to inform you that your manuscript has been judged scientifically suitable for publication and will be formally accepted for publication once it meets all outstanding technical requirements.

Kind regards,

Iddya Karunasagar

Academic Editor

PLOS ONE

Additional Editor Comments (optional):

All reviewer comments have been addressed satisfactorily.

Reviewers' comments:

Reviewer's Responses to Questions

**Comments to the Author**

1. If the authors have adequately addressed your comments raised in a previous round of review and you feel that this manuscript is now acceptable for publication, you may indicate that here to bypass the “Comments to the Author” section, enter your conflict of interest statement in the “Confidential to Editor” section, and submit your "Accept" recommendation.

Reviewer #1: All comments have been addressed

2. Is the manuscript technically sound, and do the data support the conclusions?

Reviewer #1: Yes

3. Has the statistical analysis been performed appropriately and rigorously? 

Reviewer #1: Yes

4. Have the authors made all data underlying the findings in their manuscript fully available?

Reviewer #1: Yes

5. Is the manuscript presented in an intelligible fashion and written in standard English?

Reviewer #1: Yes

6. Review Comments to the Author

Reviewer #1: The comments to the reviewer's queries have been adequately responded.and ncessary changes have been made in the manuscript

7. PLOS authors have the option to publish the peer review history of their article (what does this mean?). If published, this will include your full peer review and any attached files.

Reviewer #1: No

---

## [Editor Report · Acceptance letter]

2 Jan 2023

PONE-D-22-18648R2 

Risk factors and microbiological features of recurrent *Escherichia coli* bloodstream infections 

Dear Dr. Park:

I'm pleased to inform you that your manuscript has been deemed suitable for publication in PLOS ONE. Congratulations! Your manuscript is now with our production department. 

Kind regards, 

on behalf of

Dr. Iddya Karunasagar 

Academic Editor

PLOS ONE